# Friend or Foe: Regulation, Downstream Effectors of RRAD in Cancer

**DOI:** 10.3390/biom13030477

**Published:** 2023-03-05

**Authors:** Zhangyue Sun, Yongkang Li, Xiaolu Tan, Wanyi Liu, Xinglin He, Deyuan Pan, Enmin Li, Liyan Xu, Lin Long

**Affiliations:** 1Department of Biochemistry and Molecular Biology, Shantou University Medical College, Shantou 515041, China; 2Cancer Research Center, Institute of Basic Medical Science, Shantou University Medical College, Shantou 515041, China; 3Guangdong Provincial Key Laboratory of Infectious Diseases and Molecular Immunopathology, Shantou University Medical College, Shantou 515041, China; 4The Key Laboratory of Molecular Biology for High Cancer Incidence Coastal Chaoshan Area, Shantou University Medical College, Shantou 515041, China; 5Institute of Oncologic Pathology, Shantou University Medical College, Shantou 515041, China

**Keywords:** RRAD, dual identity, regulation, downstream effectors, cancer

## Abstract

Ras-related associated with diabetes (RRAD), a member of the Ras-related GTPase superfamily, is primarily a cytosolic protein that actives in the plasma membrane. RRAD is highly expressed in type 2 diabetes patients and as a biomarker of congestive heart failure. Mounting evidence showed that RRAD is important for the progression and metastasis of tumor cells, which play opposite roles as an oncogene or tumor suppressor gene depending on cancer and cell type. These findings are of great significance, especially given that relevant molecular mechanisms are being discovered. Being regulated in various pathways, RRAD plays wide spectrum cellular activity including tumor cell division, motility, apoptosis, and energy metabolism by modulating tumor-related gene expression and interacting with multiple downstream effectors. Additionally, RRAD in senescence may contribute to its role in cancer. Despite the twofold characters of RRAD, targeted therapies are becoming a potential therapeutic strategy to combat cancers. This review will discuss the dual identity of RRAD in specific cancer type, provides an overview of the regulation and downstream effectors of RRAD to offer valuable insights for readers, explore the intracellular role of RRAD in cancer, and give a reference for future mechanistic studies.

## 1. Introduction 

Ras-related associated with diabetes (RRAD, RAD) is a member of small GTP binding family of proteins within the Ras superfamily, sometimes referred to as the RGK family, which was first identified in type II diabetes patients because of its abnormally high expression [1,2]. As a member of Ras-related GTPase superfamily, RRAD functions as a molecular switch through modulating GTP/GDP exchange and the intrinsic GTPase activity. The human RRAD gene maps to chromosome 16q22.1 and encodes a 308 amino acid protein (Figure 1a) [3]. Conserved structural features distinguish it from other members of the RGK family include Gem, Rem1, and Rem2 [4,5]. The Ras-related GTPases have been implicated in a wide spectrum of cellular functions, including cell proliferation/differentiation [6], intracellular vesicular trafficking [7], and cytoskeletal control [8]. The basic structure of RGK proteins consists of a Ras-related core, a non CAAX–containing COOH-terminal extension, and NH2 terminal extensions [2]. The RRAD exhibits all five of the highly conserved GTPase domains (G1 to G5) that are postulated to play catalytic functions of the Ras-related protein superfamily [9] but differ from archetypical RAS at key residues involved in nucleotide binding and cycling, most notably in the G3/Switch II region [10]. The G2 domain of RRAD, which is responsible for GTPase-activating protein (GAP) binding by the p21-Ras family members, is quite different from that observed for N-Ras, Rap2b, and Ral [1]. The sequence alignment of human RRAD and the other Ras related GTPases are carried out using Clustal Omega (Figure 1b). The RRAD is longer at both the NH2 terminus (88 amino acids) and the COOH terminus (31 amino acids) than most other members of the Ras family. It is primarily a cytosolic protein that associates with the cytoskeleton and membrane fraction [11] in a nonlipid dependent manner because of the lack of a prenylation motif. The C-terminal residues 278–297 of human Rad is crucial for localization corresponding to the CaM-binding domain of RRAD. Additionally, interaction of RRAD and CaM is dependent on the guanine nucleotide bound state of RRAD [12].

Generally, the current studies on intracellular function of RRAD have been limited to specific cancer type. Tissue type plays a larger role in cancer genetics than previously realized, for example, drugs that target the drivers of proliferation will have variable efficacy depending on where the cancer is located. RRAD is found to play opposite roles as a tumor suppressor gene or oncogene in human cancers, depending on cancer and cell type. There has been no systematic summary based on the dual role of RRAD in cancer genesis, which would allow them to roundly review the position, function, and mechanism of RRAD.

We reviewed 130 published papers related to this topic and provide a glimpse into the mechanism of tissue specificity in tumorigenesis of RRAD (Figure 2). This review mainly summarizes the regulation and downstream effectors of RRAD as its dual identities in cancer genesis (Table 1 and Table 2), as well as the oncogenesis role of RRAD in senescence and current therapeutic strategy which potentially targets RRAD, thus providing a reference for future studies on the mechanisms and therapeutic developments.

## 2. Regulation of RRAD

### 2.1. DNA Methylation

DNA methylation within the promoter of genes, as one of the main epigenetic mechanisms, leads to decreased gene expression (also known as gene silencing). Accumulating evidence suggests that the RRAD promoter is hypermethylated in human cancers, such as nasopharyngeal carcinoma [17,18], breast cancer [26], esophageal cancer [30], glioblastoma [42], cervical carcinoma [28], ovary cancer [27,31], and lung cancer [24,26]. The promoter hypermethylation is associated with reduced RRAD expression in tumor tissues. In ovarian cancer, RRAD hypermethylation is mediated by RasV12 oncogenic transformation [31]. Ras is a molecular switch that mediates signal transduction across the membrane, whose dysregulation ultimately leads to oncogenesis [43]. Ras mutation is not the only cause for RRAD epigenetic inactivation in cancer, since it is found only in a certain subset of ovarian cancers [44] and 19–20% of other human cancers [43,45]. Using a well-defined ovarian cancer model which was demonstrated with hypermethylated promoter, Wang, li et al. [31] found that activation of the Ras pathway upregulates the expression of DNA methyltransferases (DNMTs), and Ras-responsive transcription factors (RRTF) may recruit DNMTs to the RRAD promoter to induce DNA methylation. As RRAD expression is aberrant in different senescence inducing conditions [37,46], whether RRAD expression is regulated by senescence-related transcription factor, C/EBP-β, which cooperates with Rb-E2F axis in RasV12-induced cellular senescence [47], needs further investigation.

### 2.2. Histone Demethylation

RRAD is epigenetic silenced in cancer by histone demethylation. Pseudogene DUXAP10 is significantly upregulated in 93 human NSCLC tissues and cell lines, which are associated with patients’ poorer prognoses and short survival time. As a key downstream mediator of DUXAP10, RRAD is epigenetic silenced through the interaction of DUXAP10 and LSD1 in NSCLC cells [19]. LSD1 alone demethylates H3K4me1/2 in peptides or bulk histones, but not in nucleosomes [48].

### 2.3. Transcription Factors Alteration

RRAD epigenetic silencing in cancer depends on more than changes in DNA methylation. BRG1/SMARCA4 is the only one of the two mutually exclusive ATPases in SWI/SNF chromatin-remodeling complex, which uses the energy of ATP hydrolysis to mobilize nucleosomes and remodel chromatin [49,50,51]. Loss of BRG1(SMARCA4) contributes to RRAD silencing epigenetically during NSCLC development [23]. Potential mechanism of gene silencing associated with the SWI/SNF complex could firstly involve altered activity of transcription factors. Previous studies have shown interactions between various SWI/SNF complex members and transcription factors, including c-MYC, NRF2, p53, and NOX [52,53,54,55,56]. Lee, D. et al. show that transactivation function of p53 is stimulated by overexpression BRG1 and dominant forms of BRG1 repress p53-dependent transcription [53]. As RRAD is a direct transcriptional target of p53, RRAD may be regulated by altered p53 related to BRG-1 mutation. Secondly, it could involve changes in nucleosome positioning after loss of BRG1 expression [52,57], and thirdly, it could associate with different histone-modifying enzymes. For example, the SWI/SNF complex can cooperate with histone acetyl transferases to promote epigenetic marks at histones and coactivator-associated arginine methyltransferase-1 to increase activity for histone methylation [58,59].

RRAD silence results from transcription factor inactivation. RRAD is a direct transcriptional target of p53 and NF-κB, and the activation of p53/NF-κB leads to upregulation of RRAD [24,60]. Multiple studies have demonstrated the change of RRAD expression level when p53 or p100 (a member of NF-κB family) is deficient. Amino acid 47 of p53 mutate from proline to serine (S47 variant) is an intrinsically poorer tumor suppressor and RRAD was downregulated in S47 tumor cells [61,62]. HPV16 E6 and E7 proteins were the main oncogenes in chronic infection associated with lung cancers, which play function by inhibiting RRAD in lung cancer cells [20]. As E6 proteins inhibited cell apoptosis mainly by degrading p53 gene [63], low RRAD expression level may result from p53 inactivation. E7 promoted cell proliferation mainly by inhibiting Rb phosphorylation [64]. As RRAD can directly bind to Grap2 and cyclin D interacting protein (GCIP), which has the ability to reduce phosphorylation of Rb, RRAD may regulate proliferation through its interaction with GCIP [60]. The detailed relationship between E7 protein and RRAD remains to be further explored. Deregulated p100 processing has been associated with constitutive NF-κB activation in breast and non-small-cell lung cancer (NSCLC) cell lines [65,66]. The expression of RRAD is enhanced in spleens of p100-deficiency mice, which suggests that incorrect regulation of the alternative NF-κB pathway strongly influences the expression of TNF target genes/inflammatory genes, such as RRAD [67]. In addition, radiation-enhanced binding of both p53 and RelA (coding the p65 NF-κB subunit) has been observed in a putative regulatory region of the RRAD gene whose expression is downregulated both by p53 and RelA silencing [68].

Previous work identified RRAD as an early growth response protein 1/2 (EGR1/2) target gene, and EGR expression levels are positively related to RRAD expression levels in prostate cancer [40]. Corepressors NAB1/2 (NGFI-A binding protein) interact with and repress EGR1 (also called NGFI-A/zif268), which play an important role in various cancer development [69,70,71,72]. EGR1 is overexpressed in prostate cancer, while NAB2 expression is reduced in most prostate cancer samples [73], which suggest that repression of EGR1 activity promote prostate cancer. RRAD is a target gene of EGR1. The EGR1-responsive regions located at −62 and −74 bp of RRAD promoter and NAB can bind with EGR1 to repress PDGF induced RRAD upregulation in a dose- and time-dependent manner [74]. Platelet-Derived Growth Factor (PDGF), acting as vascular endothelial growth factor, plays a pivotal role in regulating tumor growth and metastasis by targeting malignant cells, vascular cells, and stromal cells [75,76], which enhances EGR1 binding to RRAD promoter [74]. This observation is consistent with the idea that upregulation of RRAD is a progression factor in prostate cancer. Interestingly, EGR1 promoter contains a functional NF-κB (p65/RelA) binding site and RRAD inhibits the nuclear translocation of p65/RelA [20]. EGR1/RRAD/NF-κB axis may form a negative feedback loop, that is, EGR-1 activation induced RRAD expression and by inhibiting NF-κB binding. RRAD further inhibited transcription of EGR1 in cancer genesis.

NAB2 and CHD4 colocalize to an EGR2 binding site 90 bp upstream of the start site within the RRAD promoter. To repress activation of EGR2 target RRAD transcription, CHD4-interacting domain (CID) of NAB2 recruit the nucleosome remodeling and deacetylase (NuRD) complex and associate with its chromodomain helicase DNA-binding protein 4 (CHD4 subunit). The repression through the CID is dependent on histone deacetylases (HDAC), subunits of the NuRD complex, which impede dissociation of DNA from histone octamer [77].

## 3. Diverse Downstream Effectors of RRAD

### 3.1. RRAD as Tumor Suppressor Gene 

The role of RRAD is not necessarily independent; the regulation and downstream targets of RRAD as tumor suppressor gene may interact with each other (Figure 3).

#### 3.1.1. Cell Signaling

RRAD can negatively regulate TNFα-stimulated NF-κB pathway which promotes GLUT1 translocation. In lung cancer, RRAD directly bounds to the subunit p65 in the NF-κB complex and negatively regulates the activation of NF-κB by inhibiting p65 translocate to the nucleus [21,42]. The mechanism of p65 subunit binding with RRAD export nucleus may be the interaction of RRAD with RelA/p65, and might somehow favor the association with IκB. It is believed that the nuclear export signals (NES) are in IκB, and the nuclear localization signals (NLS) are in the NF-κB subunits [78,79]. However, it could also be that RRAD may regulate NF-κB localization through its effects on the Rho/ROCK/Cofilin/actin axis. As aforementioned, RRAD can modulate a Rho signaling pathway by binding to ROCK and interfere the phosphorylation of cofilin. In addition, both RRAD and RelA/p65 [80] have been reported to interact with the 14-3-3 scaffolding protein which is known to bind and retain specific cellular proteins in the cytoplasm.

Inhibiting the phosphorylation of p65 subunit in the NF-κB complex can also negatively regulate the activation of NF-κB [81]. In a recent study, Na-Jin Gu et al. found that inhibiting RRAD significantly increased the nuclear translocation of p65 and the expression level of p-p65 [20] which leads to abnormal activation of NF-κB pathway. NF-κB controlled transcription in a gene-specific manner [82]. Hypoxia-inducible factor-1α(HIF-1α) is a downstream target gene of NF-κB in esophageal cancer [83], breast cancer [84], and hepatic cancer [85]. Downregulation of RRAD in lung cancer leads to increased TNFα-stimulated NF-κB target genes such as MMP9 [21], HIF-1α, and GLUT1 at both protein and mRNA levels [20]. In hepatocarcinoma cells, knockdown of RRAD not only increases the expression of GLUT1 but also promotes the expression of hexokinase II(HK-II) [16]. Overexpression of MMP9 has been reported in different types of cancer and it is believed to facilitate tumor invasion and metastasis [86,87]. HK-II is a key rate-limiting enzyme in glycolysis which has been found to be overexpressed in many tumor tissues and accelerated the tumor aerobic glycolysis [88,89]. HIF-1α is a transcriptional complex which has a key role in regulation of genes involved in energy metabolism, angiogenesis, and apoptosis [90]. When HIF-1α dimer is formed and enters the nucleus, it binds to the hypoxia response element (HRE) and stimulates GLUT1, HK-II transcription [91].

#### 3.1.2. Cancer Cell Proliferation

RRAD lowers cell proliferation, arrests the cell cycle, and increases apoptosis by binding and downregulating ACTG1 which acts as a functional downstream effector of RRAD in HCC cells. Expression of RRAD is low when ACTG1 is overexpressed in HCC tumor specimens and is linked to poor prognosis [13]. ACTG1 promoted HCC proliferation in several ways: (1) regulating the cell cycle via increasing expression of cyclin A2, cyclin D1, cyclin E1, CDK2, and CDK4 which results in cell cycle transition in G0/G1 [92,93]; (2) inhibiting mitochondrial apoptosis pathway by decreasing Bax and cleaved poly(ADP-ribose) polymerase/caspase-3(PARP) [94,95]; and (3) upregulating the GLUT1 expression [15]. The role of RRAD is not necessarily independent; the regulation and downstream targets of RRAD as a tumor suppressor gene may interact with each other.

#### 3.1.3. Cancer Cell Migration

RRAD disturbs cancer cell migration through Rho signaling pathway by binding to 14-3-3 and ROCK (Rho-associated protein kinase), thus interfering in the abundance of pSer3-cofilin that is a regulator of actin dynamics in NSCLC cells [24]. ROCK is a major effector of Rho GTPases. Members of the Rho family GTPase are key regulators of actin cytoskeleton and play important roles in cellular processes such as cell morphogenesis and movement [96]. RRAD binds and inhibits ROCKII [2], functioning as a tumor suppressor gene. In addition, RRAD interacts with 14-3-3 scaffold proteins and alters its subcellular localization [97], and 14-3-3 protein negatively regulates pSer3-cofilin levels by modulating the activities of SSH1L phosphatase, which dephosphorylates pSer3-cofilin [98]. RRAD competes for SSH1L binding to 14-3-3, relieving its inhibition from 14-3-3, thus inhibiting the cell migration.

#### 3.1.4. Energy Utilization

RRAD decreases energy utilization efficiency of cancer by inhibiting the Warburg effect. Altered metabolism includes increased glucose uptake, and fermentation of glucose to lactate is observed even in the presence of completely functioning mitochondria, which is known as the Warburg Effect [99]. It has been postulated that proliferating cancer cells could choose glycolysis to efficiently prepare nutrients for fast cell growth [100,101]. Previous studies have shown that RRAD is associated with and phosphorylated by protein kinases including calmodulin-dependent protein kinase II, cAMP-dependent protein kinase (PKA) and protein kinase C (PKC) [102,103]. Activation of PKA and PKC increased the activity of the Na-glucose cotransporter SGLT2 [104], and calmodulin-dependent protein kinase II positively regulates glucose uptake by affecting the GLUT4 (glucose transporter 4) expression level [105,106]. Since most of these protein kinases are involved in glucose metabolism, RRAD is speculated to act as the downstream for these protein kinases and repress glucose uptake by inhibiting the expression or activities of glucose transporters. Glucose transporters (GLUTs) mediated glucose across the plasma membrane (PM) of cells which is the first rate-limiting step for glucose metabolism [107]. GLUT4 is the main glucose transporter expressed in insulin-responsive fat and muscle tissues, being translocated to the PM to promote the glucose uptake. Although RRAD displays a negative regulator of glucose uptake in cultured muscle and fat cells, this is due to a decrease in the intrinsic activity of the transporter molecules, rather than an effect on the translocation of GLUT4 [11]. GLUT1 is ubiquitously expressed in various cells and tissues and is responsible for constitutive glucose uptake [107]. Overexpression of RRAD inhibits the GLUT1 translocation to the PM, which is an important mechanism of RRAD to repress the aerobic glycolysis in cancer cells [22].

### 3.2. RRAD as Oncogene 

As oncogene, RRAD participates in various molecular pathways and promotes cancer progression by facilitating cell proliferation and migration (Figure 4).

#### 3.2.1. Cell Signaling

RRAD promotes malignant glioma progression via endosome-mediated epidermal growth factor receptor (EGFR)/STAT3 signaling [34]. Although primary glioblastoma tumors are reported to express significantly low levels of RAS transcripts and no detectable levels of RAS proteins [108], analysis of gene expression of human glioma tissue samples deposited in the REMBRANDT database clearly implicates a correlation between upregulation of RRAD in EGFR-expressing glioma patients and poorer prognosis [34]. Constitutively activated STAT3 is frequently co-expressed with EGFR in high-grade gliomas [109] and cooperates with EGFR to facilitate epithelial-mesenchymal transition in human epithelial cancers [110]. EGFR family is a member of RTKs, which has been shown to localize to the nucleus [111,112]. Through the nuclear pore complex, EGFR translocate from the cell surface to the inner nuclear membrane, which is mediated by RRAD associated importin-β with three conserved NLS regions [34,97,113]. Besides, Yeom, Nam et al. found that the membrane-bound early endosomal marker, EEA1, also coprecipitated with RRAD. RRAD enhances EGFR protein stability which induced STAT3 phosphorylation. Phosphorylated STAT3 colocalizes with receptor–ligand complexes on the endosome and is transported from the plasma membrane to the perinuclear region [114], which enhances the expression of several downstream genes including EMT-regulating proteins and stemness-regulating transcription factors. EMT-regulating proteins (TWIST, SNAIL, and SLUG) [34] are upregulated, which promotes tumor invasion and metastasis [115]. Stemness-regulating transcription factor (OCT4, NANOG, and SOX2) levels increased with RRAD overexpression [34], which are critical for maintaining self-renewal, proliferation, survival, and multilineage differentiation potential of GBM stem cells [116]. RRAD inhibition could also suppress expression of EMT-associated proteins (VIMENTIN, TWIST, SNAIL, and OCCLUDIN), VEGF, and ANGP2 in gastric cancer (GC) and colorectal cancer (CRC) cell lines [39]. VEGF is a positive regulator of tumor angiogenesis, and VEGF inhibitors are widely used in cancer treatment [117]. ANGP2 is also involved in angiogenesis of tumor tissue [118]. This observation indicates the oncogene effect of RRAD in tumor progression.

#### 3.2.2. Cancer Cell Proliferation and Migration

RRAD modulates cancer cells’ proliferation and motility by directly interacting with CaM, CaMKII [12], and β-tropomyosin [119]. Overexpression of RRAD in the breast cancer cells results in acceleration of cell cycle transition which is dependent on its NH2- and COOH-terminal regions [27] which contain CaM [12]. CaM can regulate the G1-S transition of the cell cycle by increasing the activities of CDK2, CDK4, and pRb [120]. Besides, RRAD interacts with CaM, CaMKII, and β-tropomyosin which can regulate the cytoskeletal organization and promote cancer cell proliferation and motility [121,122].

Furthermore, in Tseng, Vicent et al. ’s study, the function of RRAD, is blocked by the co-expression of nm23, and in addition, similar effects are also seen when these cells are injected into nude mice [27]. RRAD exhibits a novel form of bi-directional interaction with the nm23 that a putative tumor metastasis suppressor [123]. Nm23 act as both a GTPase-activating protein and a guanine nucleotide exchange factor for RRAD, determining the balance between GTP-RRAD and GDP-RRAD [41]. GDP-bound form RRAD is favored with CaM, CaMKII, and β-tropomyosin [12,119], indicating a regulator role of nm23 in cell cytoskeleton and mobility. More extensive study will be required to determine the in vivo significance in human tumors of an interaction with nm23 and the prognostic significance of RRAD expression.

RRAD binds and inhibits Grap2 and cyclin D interacting protein (GCIP), a cell cycle-inhibitory molecule, to promote carcinogenesis. Lee, Yeom et al. have demonstrated that RRAD binds directly to GCIP in vitro and coimmunoprecipitates with GCIP from cell lysates. RRAD translocates GCIP to the cytoplasm and inhibit the tumor suppressor activity of GCIP, which ultimately increases Rb phosphorylation and upregulates cyclin D1 [60].

#### 3.2.3. Energy Utilization

Although RRAD is mainly found as a negative regulator of Warburg effect, Kim, H.K. first found the level of lactate was decreased without increasing uptake of glucose when transfected with siRRAD in GC or CRC, which suggests that RRAD may be a positive regulator of the aerobic glycolysis [39]. This opposite finding remains to be investigated to further understand the role of RRAD in tumor cell energy metabolism.

### 3.3. RRAD in Senescence Contributes to Tumor Progression

Cellular senescence is a state of irreversible growth arrest in cells with a variety of stresses that could lead to transformation, and senescent cells are observed as the first barrier against tumorigenesis in precancerous lesion [124]. However, the senescence-associated secretory phenotype (SASP) has been shown to promote the tumor resistance to chemotherapy and tumor relapse in certain circumstances, instead of enforcing arrest and recruiting immune cells to contribute to suppress tumor progression as usual [125,126,127]. Successful elimination of senescent cells also takes place in tumor suppression activity [127].

RRAD is the first reported p53 target gene that functions to impede cellular senescence. It is possible that in the context of oncogenesis, RRAD downregulation may promote cellular senescence and consequently reduces apoptosis by increasing the level of reactive oxygen species (ROS) [37]. Different from alleviating oxidative stress under physiological conditions, p53 can exacerbate oxidative stress under highly stressed conditions through regulating the level of intracellular ROS, whereas RRAD upregulation may counteract this process [37], which is in the spotlight for both anti-cancer and anti-aging therapies.

RRAD may also restrain senescence by inhibiting the function of NF-κB that usually upregulates in senescent cells and promotes senescence. As senescent cells also favor glycolysis for the high-efficient output of biosynthetic precursors similar with tumor cells [128]. RRAD limits glycolysis through inhibiting GLUT1 translocation to the plasma membrane by negatively regulating NF-κB [22,31,42], and the transactivation of RRAD by NF-κB may represent a negative feedback mechanism to restrain senescence [37]. Interestingly, p53 and NF-κB promote RRAD expression together in cellular senescence [37].

In addition, cell senescence is associated with cell cycle inhibitors [129]. Rb phosphorylation is critical for both the cell cycle and senescence [130]. The RRAD knockdown increased the level of p27 and decreased Rb phosphorylation, which may depend on binding with GCIP to decrease GCIP-induced Rb phosphorylation downregulation [60]. Furthermore, pRb causes a posttranscriptional accumulation of the cyclin dependent kinase inhibitor p27KIP1, which inhibits the activity of cyclin E kinase to trigger senescence [129].

RRAD hypermethylation is mediated by RasV12 oncogenic transformation [31]. Whether RRAD expression is regulated by other senescence-related transcription factor, such as C/EBP-β, which cooperates with RB-E2F axis in RasV12-induced cellular senescence [47], needs further investigation.

### 3.4. RRAD as a Potential Drug Target

In view of its pro-tumorigenic role in cancer, RRAD may serve as a promising target for therapeutic intervention. Based on screening analyses, five classes of drugs were determined as potent inhibitors of RRAD-expressing glioblastoma, including antitussive agents(oxeladin), anthelmintics (AH), microtubule inhibitors (MI), and topoisomerase inhibitors (TI). However, owing to the nonspecific cytotoxicity (MI/TI) and efficacy (AH) against cancers, Lee, Yeom demonstrated that RRAD inhibition results from the hydrogen bonding between oxeladin and the Gln250 of RRAD, and the pi-pi stacking interactions of oxeladin with the Lys228 or Arg249 of RRAD [35]. Components of RRAD-associated signaling cascades, including pEGFR, pAKT, pERK, and pSTAT3, were inhibited in the presence of oxeladin. Besides, RRAD promotes malignant progression and enhances resistance to temozolomide via endosome-mediated EGFR/STAT3 signaling [34]. Given that RRAD knockdown sensitizes chemo-resistant cancer cells to cytotoxic drugs, targeting STAT3 with oxeladin or butamirate may attenuate temozolomide resistance and glioblastoma recurrence after treatment [38,60]. RRAD is over expressed in leukemia/lymphoma cell lines. The knockdown of RRAD diminished the survival of bortezomib-resistant cancer cells by inducing mitochondrial apoptosis via proapoptotic Noxa/Bcl-2 modulation, which might be caused by caspase activation. Besides, treatment with the PI3K inhibitor induced downregulation of p-Akt (Ser473) and RRAD. RRAD activates Akt and inhibition of the Akt pathway can inhibit bortezomib resistance. Collectively, these findings indicate the critical involvement of RRAD in bortezomib resistance [38]. Additionally, taxanes (paclitaxel) showed synergism with RRAD inhibition in CRC cell lines, which support the feasibility of an RRAD inhibitor as a therapeutic target for treatment of GC and CRC [39].

## 4. Conclusions and Perspectives

RRAD functions as an oncogene or tumor suppressor gene depends on different cancer. The expression level of RRAD can be modulated epigenetically and by multiple transcription factors, which is associated with cancer progression. In addition, RRAD interacts with diverse downstream effectors to play the dual roles in cancer genesis. It can regulate cell cycle transition, cytoskeletal stability, cell motility, and the tumor micro-environment composition, thus influencing tumor progression and metastasis.

In conclusion, by reviewing the intracellular role of RRAD in cancer genesis, we summarized the detailed function of RRAD in specific cancer and cell type, providing an overview of regulation mode and downstream targets of RRAD in both tumor suppressing and oncogenic aspect. We hope that these descriptions can help readers further understand the role of RRAD in cancer cell progression, explore the genetic changes that underlie human cancer, and provide practical ideas for targeted therapy for the disease.

## Figures and Tables

**Figure 1 biomolecules-13-00477-f001:**
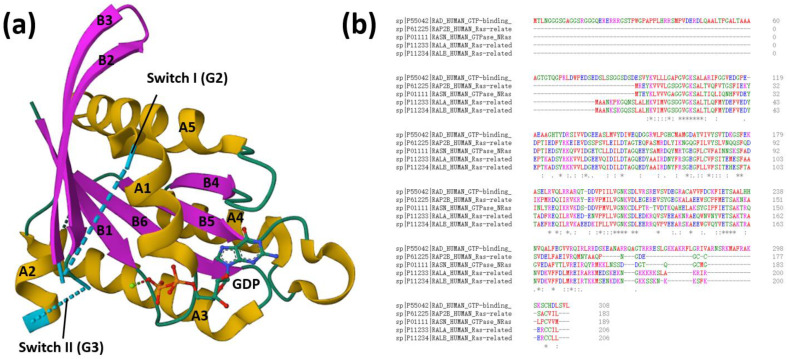
Structure of human RRAD. (**a**) The crystal structures of human RRAD. RRAD contains five α-helices (A1–A5) in yellow and one large β-sheet in pink (PDB ID: 2DPX, general view), that comprise two extended anti-parallel β-strands (B2 and B3) and five extended parallel β-strands (B3, B1, B4–B6). Both Switch I and Switch II of RRAD are disordered. The switch I region of RRAD lacks the G2 motif which is generally contained by small GTPases. This motif is important for nucleotide binding. Additionally, RRAD lacks the conserved phenylalanine (Phe28 of H-Ras) that traps the GDP molecule through non-polar interactions with the base and sugar. However, Arg109 on helix A1 serves to hold the nucleotide by forming a water-mediated hydrogen bonding network. The magnesium ion is combined with the side chain of Ser105. As Ser105 mutation can eliminate the binding of GTP, magnesium ion seems to be more critical to the binding of GDP and Rad than other GTP enzymes. The disordered switch II suggests conformational flexibility. The G3 motif comprises the N-terminal half of switch II. RRAD displays no significant changes in the overall fold compared to other Ras-family GTPases. (**b**) Sequence alignment of human RAD, RAP2B, RASN, RalA, and RalB.(https://www.ebi.ac.uk/Tools/services/web/toolresult.ebi?jobId=clustalo-I20230227-084742-0410-26884510-p1m&showColors=true&tool=clustalo, accessed on 27 February 2023). (*) indicate strict identity between sequences; (:) indicate strong similarity; (.) indicate weak similarity and (-) represent alignment gaps.

**Figure 2 biomolecules-13-00477-f002:**
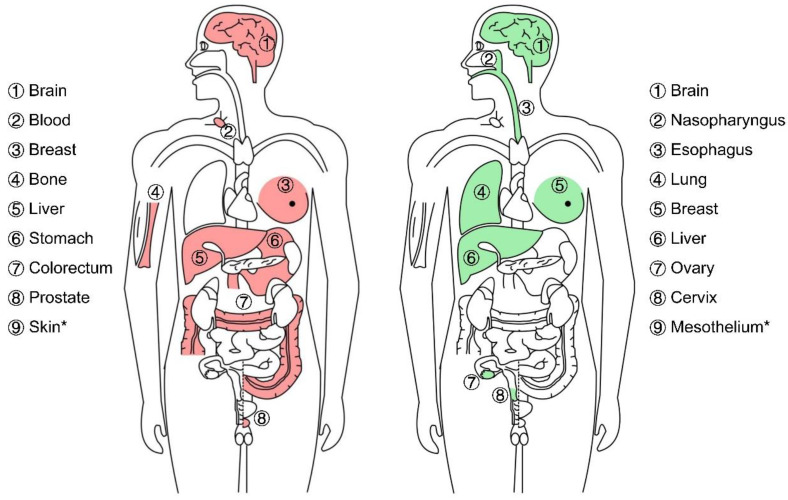
Dual role of RRAD in specific cancer types. Red represents oncogene effect of RRAD, whereas green represents tumor suppressor effect. Cancers with * are not represented in the figure. Detailed explanations are summarized in Table 1 and Table 2.

**Figure 3 biomolecules-13-00477-f003:**
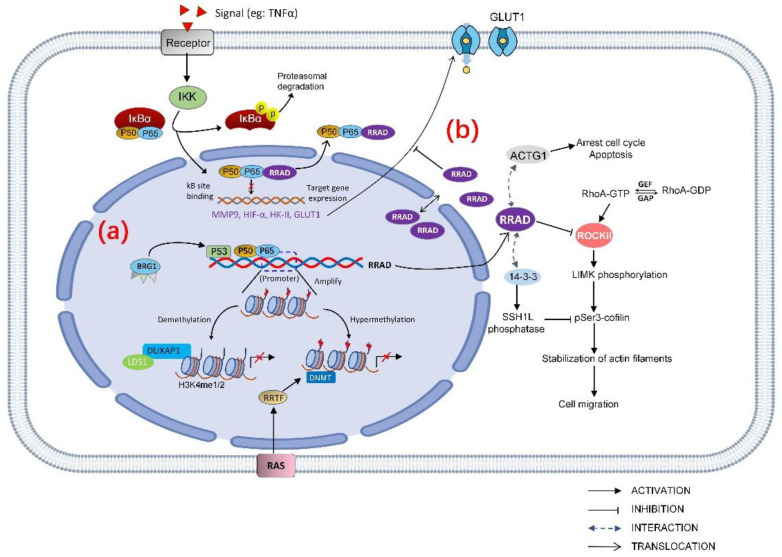
Regulation and downstream effectors of RRAD as a tumor suppressor gene. (**a**) RRAD regulation can be categorized as hypermethylation of the RRAD promoter, demethylation of the RRAD promoter region, and transcription factors modulated the RRAD gene, all of which eventually led to reduced RRAD expression. p53 and NF-κB are transcription factors directly target RRAD. Detailed explanations are summarized in Table 1. (**b**) RRAD has various downstream effectors. RRAD directly binds to the subunit p65 in the NF-κB complex, negatively regulates the activation of NF-κB by inhibiting the p65 translocation to the nucleus. RRAD inhibits the GLUT1 translocation to PM, which suppresses aerobic glycolysis in cancer cells. In addition, RRAD disturbs cancer cell migration through Rho signaling pathway by interacting with 14-3-3 and inhibits ROCK (Rho-associated protein kinase); RRAD arrests the cell cycle and increases apoptosis by regulating ACTG1.

**Figure 4 biomolecules-13-00477-f004:**
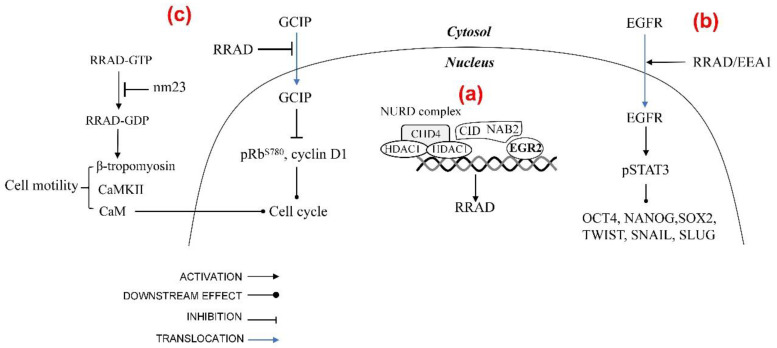
Regulation and downstream effectors of RRAD as an oncogene. (**a**) RRAD is EGR target gene. Corepressors NAB reduced in cancer, which interact with and derepress EGR1 to upregulate RRAD expression. NAB2 repress activation of EGR2 by recruiting the nucleosome remodeling and deacetylase (NuRD) complex. (**b**) RRAD coprecipitated with EEA1 promotes endosome-mediated EGFR translocation. RRAD enhances EGFR protein stability, which induced STAT3 phosphorylation, which enhances the expression of TWIST, SNAIL, SLUG, OCT4, NANOG, and SOX2. (**c**) The balance between GTP-RRAD and GDP-RRAD is regulated by nm23, coexpression of which blocks the function of RRAD. GDP-RRAD is favored with CaM, CaMKII, and β-tropomyosin that enhance cell motility. CaM can also control the G1-S transition of the cellular cycle. RRAD translocates GCIP to the cytoplasm and inhibits GCIP, which ultimately promote cell cycle transition.

**Table 1 biomolecules-13-00477-t001:** Tumor suppressor effect of RRAD.

Tumor Type	Models	Pathway	Function	Refs.
Liver	Hepatic cancer patient specimen	P53/RRAD/ACTG1	low expression levels of RRAD are significantly correlated to large tumor size, advanced tumor stage and bad prognosis.	[13]
	Human hepatocellular carcinoma cell lines			
	Nude mice			
	Wild type and S47 E1A/RAS transformed mouse embryo fibroblast lines	P53/RRAD/GLUT 1	Low RRAD expression levels significantly increase GLUT1 in S47 tumor cells with increased risk for hepatocellular carcinoma and other cancers.	[14]
	Human hepatocellular carcinoma cell lines		RRAD inhibits cell proliferation and migration, arrests the cell cycle and increases apoptosis.	[15]
	Hepatic cancer patient specimen			
	nude mice			
	Human hepatocellular carcinoma cell lines	RRAD/GLUT 1, HK-II	low expression of RRAD was associated with tumor size, microvascular invasion or metastasis, and tumor node metastasis (TNM) stage.	[16]
	Hepatic cancer patient specimen			
Nasopharynx	NPC patient specimen	Hypermethylation	RRAD is hypermethylated in nasopharyngeal carcinoma cell. Low expression levels of RRAD induce proliferation, colony formation, and migration.	[17,18]
	NPC cell lines			
Lung	NSCLC patient specimen	DUXAP10/ LSD1/RRAD	DUXAP10 downregulates RRAD through binding with LSD1 could promote the cell cycle progression and proliferation phenotype of NSCLC cells.	[19]
	NSCLC cell lines			
	Lung cancer cell lines	HPV/RRAD/NF-Κb/HIF-1α/GLUT 1	RRAD downregulates the expression of both HIF-1α and GLUT1.	[20]
	Lung adenocarcinoma cell lines	RRAD/ NF-κB/ MMP9	RRAD suppresses MMP9 expression and cell invasion.	[21]
	Lung cancer cell lines	P53/RRAD/GLUT1	RRAD greatly reduces glycolysis through inhibition of GLUT1 translocation to the plasma membrane in lung cancer cells.	[22]
	NSCLC cell lines Adrenal carcinoma cell lines	BRG1(SMARCA4)/ /RRAD	RRAD expression levels decrease in BRG1 mutation NSCLC cells.	[23]
	NSCLC patient specimen	P53/RRAD/ROCK2/LIMK/ cofilin/actin P53/RRAD/14-3-3/SSH-L1/cofilin/actin Methylation	RRAD promotes the stabilization of actin, inhibit cell migration.	[24]
	NSCLC cell lines			
	31 lung cancer cell lines (20 NSCLC and 11 SCLC cell lines) Non-malignant human bronchial epithelial cells (NHBEC)	Methylation	RRAD methylation suppresses its transcription and affect survival of patients with lung cancer.	[25]
Breast	breast cancer cell lines	Methylation	RRAD gene is hypermethylated and silenced in breast cancer patients.	[26]
	nonmalignant human mammary epithelial cells (NHMEC)			
	Breast cancer patient specimen		RRAD is frequently downregulated in non-advanced breast cancers	[27]
Cervix	Cervical carcinoma cell lines	Methylation	RRAD gene is hypermethylated and silenced in cervical cancer patients.	[28]
	Cervical patient specimen (with or without HPV)			
Brain	Glioblastoma patient specimen	Hypermethylation	RRAD gene is hypermethylated and silenced in GBM patients and associated with short-term survival.	[29]
Esophagus	Esophageal carcinoma patient specimens	Hypermethylation	aberrant methylation of RRAD may be involved in pathogenesis of a subset of ESCC.	[30]
	Esophageal adenoma patient specimens			
Ovary	Ovarian cancer cell lines	Ras V12/RRAD	downregulation of RRAD led to an increase in glucose uptake, which may be associated with cancer cell Aerobic glycolysis.	[31]
	Epithelial ovarian cancer cell lines	P53/RRAD	RRAD mRNA level is increased after aspirin acetylates p53.	[32]
Mesothelium	Malignant mesothelioma cell lines	Hypermethylation	aberrant methylation of RRAD plays a role in the pathogenesis of MM.	[33]

**Table 2 biomolecules-13-00477-t002:** Oncogene effect of RRAD.

Tumor Type	Models	Pathways	Function	Ref.
Brain	Human Glioblastoma cell lines	RTKs(EGFR)/STAT3/RRAD/EEA1	depletion of RRAD leads to decreased proliferation and survival of GBM cells RRAD is associated with chemotherapy (temozolomide) resistance. RRAD enhances self-renewing ability, tumor sphere formation, EMT, and in vivo tumorigenesis.	[34]
	Human Glioblastoma multiforme cell lines Nude mice	EGFR/STAT3	Oxelaidin targets RRAD and inhibits EGFR/STAT3 signaling pathway to influence growth, apoptosis, and chemo-resistance of glioblastoma cell lines.	[35]
Liver	HCC cell lines (derived from hepatocarcinoma patient specimen)		RRAD expression levels are upregulated in HCC cells and have capacities to promote proliferation and migration.	[36]
Bone	Human osteosarcoma cells--U2OS	(p53, NF-κB)/RRAD	RRAD knockdown resulted in increased cellular senescence in U2SO cells.	[37]
Blood	Human leukemia cell lines	PI3K/Akt/Noxa/Bcl-2	RRAD promotes the bortezomib/drug resistance.	[38]
	Lymphoma cell line		RRAD enhances resistance to bortezomib-induced apoptosis.	
Stomach	Gastric cancer cell lines Gastric cancer patient-derived cells Matched pairs of primary cancer tissue and non-tumor tissue	RRAD/Vimentin, twist, snail, and occludin; RRAD/VEGF, ANGP	RRAD promotes gastric tumor cell proliferation, invasion, EMT and angiogenesis	[39]
Colorectum	Colorectal cancer cell lines Colorectal cancer patient-derived cells Matched pairs of primary cancer tissue and non-tumor tissue	RRAD/Vimentin, twist, snail, and occludin; RRAD/VEGF, ANGP	RRAD promotes colorectal tumor cell proliferation, invasion, EMT and angiogenesis.	[39]
Breast	Breast cancer patient specimen Human breast carcinoma cell lines Nude mice		RRAD accelerates growth of breast cancer cells in vitro and increase the tumorigenicity of these cells when injected into nude mice.	[27]
Prostate	Prostate cancer cell line-PC-3 and DU145 Athymic nude mice	RRAD/GCIP	RRAD suppresses DNA damage-induced cell cycle arrest and induction of premature senescence. RRAD expression increased doxorubicin resistance. RRAD increases telomerase activity and colony formation.	[30]
	Prostate cancer cell line-LAPC4	EGR1/RRAD	RRAD is overexpressed in prostate tumors and may promote cancer cell growth as down-stream target gene of EGR1	[40]
Skin	K-1735 TK melanoma cells		RRAD can enhance DNA synthesis in response to serum.	[41]

## Data Availability

No new data were created or analyzed in this study. Data sharing is not applicable to this article.

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
