# Peer review of "Friend or Foe: Regulation, Downstream Effectors of RRAD in Cancer"

_biomolecules, 2023, doi:10.3390/biom13030477_

Round 1

Reviewer 1 Report

This review by Sun et al. is a well-written summary of the RRAD and its roles as an oncogene or tumor suppressor gene. In this review, the authors discuss published literature on the dual identity of RRAD in specific cancer type and provides an overview of the regulation and downstream effectors of RRAD to offer valuable insights. I have a few suggestions that authors should consider while revising the manuscript.

  • RRAD has also been described in the literature as RAD, so clarifying that in the beginning would be helpful.
  • The authors describe how RRAD is different from RAS proteins. It would be helpful if authors could include sequence alignment of RRAD with RAS proteins to show the extra regions present in RRAD.
  • In Protein Data Bank, there are crystal structures of RRAD/RAD, and it would be helpful if authors could add a small summary on that as well.
  • In line 91, the authors say that "30% of human cancer" is based on very old literature. A recent cancer database analysis has shown that RAS is responsible for 19-20% of all human cancer. (Ref Cancer Res (2020) 80 (14): 2969–2974.)

Reviewer 2 Report

Overall, this was a comprehensive review of the role of RRAD in cancer.  This review gave a good overview of the association of RRAD expression levels with various cancer types.  Additionally, the mechanism of RRAD gene transcription regulation was well covered. The downstream effectors of RRAD and their relationship to cancer was also expansively covered.   The content of the manuscript is generally good, however, the manuscript requires improvement before publication as detailed below.

General comments:

1. The manuscript would benefit from some restructuring. The authors have included a section titled “Oncogene Effect” and thus have two separate sections describing the “Regulation of RRAD” and the “RRAD Effectors”. It is not clear why these sections are separated as they mostly describe regulation and effectors in the context of cancer. Merging the two “Regulation of RRAD” sections would result in a more coherent section that describes both the epigenetic silencing and transcriptional activation of RRAD that is observed in cancer. Similarly the two “RRAD effector” sections, should also be merged.  These new sections could then divided into subsections based on the function they describe (Energy utilisation, signalling, proliferation/migration etc), which would improve the flow of the manuscript. Finally, the paragraph describing RRAD as a drug target (Lines 364-385) should be moved out of the conclusion to its own section preceding the conclusion paragraph.

2. The manuscript frequently misses the use of articles (eg Line 154 cancer by inhibiting the Warburg effect) and prepositions (eg Line 144 thus interfering with the abundance) and also misuses verb forms (eg Line 191: increased not increase. While the meaning of sentences is generally understandable, rectifying these issues would make the manuscript more readable.   

SPECIFIC COMMENTS

1.       Include a sentence in the first paragraph describing the molecular function of RRAD i.e that as a GTPase protein its function is to bind and catalyse the hydrolysis of GTP.

2.       Line 49, should be two sentences i.e.

“The RRAD exhibits all five of the highly conserved GTPase domains (G1 to G5) that are postulated to play catalytic functions of the Ras-related protein superfamily[8]. However, the G2 domain of RRAD, which is responsible for GTPase-activating protein (GAP) binding by the p21-Ras family members is quite different from that observed for N-Ras, Rap2b, and Ral[1]

3.       Line 60

“As tissue type plays a larger role in cancer genetics than previously realized, which also explains why drugs that target the same drivers of proliferation some times work in some cancers, and sometimes they don't.”

This sentence seems out of place here. Are you suggesting that anti-proliferative drugs have different effects in various tissues due to the presence of RRAD?  If so you should provide details. Otherwise, remove or modify this sentence: “Tissue type plays a larger role in cancer genetics than previously realized, for example drugs that target the drivers of proliferation will have variable efficacy depending on where the cancer is located”.

4.       Line 66

Were the 130 papers chosen in a systematic way i.e. all papers discovered in a pubmed search of “RRAD+Cancer”.

5.       Line 131

This is unclear and should be rewritten for example : RRAD binds directly to GCIP which has the ability to reduce phosphorylation of Rb, thus RRAD may regulate proliferation through its interaction with GCIP.

6.       Line 138

“pathway strongly influences the expression of TNF target genes/inflammatory genes--RRAD.”

What do the hyphens mean? Should this say “such as” ?

7.       Line 259

Please explain why high RRAD levels are "clearly implicated in the malignant progression of human glioblastoma."

8.       Lines 272 and 277

“twist” and “snail” have  inconsistent capitalisation.

9.       Line 323 title needs something after “tumour” – progression, perhaps?
